# Lung Function Testing and Prediction Equations in Adult Population from Maputo, Mozambique

**DOI:** 10.3390/ijerph17124535

**Published:** 2020-06-24

**Authors:** Olena Ivanova, Celso Khosa, Abhishek Bakuli, Nilesh Bhatt, Isabel Massango, Ilesh Jani, Elmar Saathoff, Michael Hoelscher, Andrea Rachow

**Affiliations:** 1Division of Infectious Diseases and Tropical Medicine, Medical Centre of the University of Munich (LMU), 80802 Munich, Germany; bakuli@lrz.uni-muenchen.de (A.B.); saathoff@lrz.uni-muenchen.de (E.S.); hoelscher@lrz.uni-muenchen.de (M.H.); Rachow@lrz.uni-muenchen.de (A.R.); 2Instituto Nacional de Saúde (INS), 3943 Maputo, Mozambique; khosacelso@gmail.com (C.K.); nbhatt.mz@gmail.com (N.B.); isabel.timana@gmail.com (I.M.); ilesh.jani@gmail.com (I.J.); 3Center for International Health—CIH LMU, 80802 Munich, Germany; 4German Centre for Infection Research (DZIF), Partner Site, 80802 Munich, Germany

**Keywords:** lung function, spirometry, prediction equation, adult, Mozambique, Africa

## Abstract

**Background:** Local spirometric prediction equations are of great importance for interpreting lung function results and deciding on the management strategies for respiratory patients, yet available data from African countries are scarce. The aim of this study was to collect lung function data using spirometry in healthy adults living in Maputo, Mozambique and to derive first spirometric prediction equations for this population. **Methods:** We applied a cross-sectional study design. Participants, who met the inclusion criteria, underwent a short interview, anthropometric measurements, and lung function testing. Different modelling approaches were followed for generating new, Mozambican, prediction equations and for comparison with the Global Lung Initiative (GLI) and South African equations. The pulmonary function performance of participants was assessed against the different reference standards. **Results:** A total of 212 males and females were recruited, from whom 155 usable spirometry results were obtained. The mean age of participants was 35.20 years (SD 10.99) and 93 of 155 (59.35%) were females. The predicted values for forced vital capacity (FVC), forced expiratory volume in 1 s (FEV1) and the FEV1/FVC ratio based on the Mozambican equations were lower than the South African—and the GLI-based predictions. **Conclusions:** This study provides first data on pulmonary function in healthy Mozambican adults and describes how they compare to GLI and South African reference values for spirometry.

## 1. Introduction

Local reference data is important for interpreting spirometry test results and deciding on the management strategies for respiratory patients. According to the American Thoracic Society (ATS) recommendations, reference values should be derived from a healthy population with the same ethnic origin and anthropometric characteristics as the participants and patients being tested in studies or clinical services [1]. The Task Force of the Global Lung Function Initiative (GLI) aimed to establish improved international lung function reference data and to derive continuous prediction equations for spirometric indices, which are applicable globally [2]. This approach included data from various countries around the world yet data from African, South Asian and Latin American countries are lacking [3]. Therefore, the reference standards published by GLI for lung function parameters of African populations may not be appropriate for use in all African settings. Local prediction equations are also unavailable in many African countries, including Mozambique. Thus, international reference values adjusted by ethnic correction factors are often used for spirometry in African settings [4,5]. This might lead to lower diagnostic standards in specific ethnic groups and, ultimately, incorrect clinical diagnoses in patients with (or without) pulmonary symptoms.

In the present study, we aimed to collect spirometry data from non-symptomatic adults living in urban Maputo, Mozambique, to derive prediction equations for this specific population, which can be used to evaluate lung function in participants of clinical research studies [6,7], and also, in patients with respiratory diseases in this area.

## 2. Materials and Methods

### 2.1. Study Design and Participants

A cross-sectional study was conducted between April and December 2017. Household members and neighborhood contacts of participants of a tuberculosis (TB) cohort study [6] conducted at the Instituto Nacional de Saúde (INS) TB Research Study Clinic in Mavalane, Maputo, and residents attending HIV counselling and testing clinic at the Mavalane Health Centre were recruited using a convenience sampling approach. Exclusion criteria were a history of TB, current symptoms of active TB, any acute or chronic respiratory diseases, and contraindications for spirometry [8,9]. Eligible participants were 18 years or older and willing to provide informed consent for study participation.

### 2.2. Data Collection

Data collection and lung function measurements were performed at the Instituto Nacional de Saúde (INS) TB Research Clinic in Maputo, Mozambique, on the premises of the Mavalane Health Centre. Height and weight measurement were taken using a stadiometer and an electronic weighing scale, respectively. Body mass index (BMI) was calculated as weight (kg) divided by height (m) squared. Demographic characteristics were collected using a short, standardized questionnaire developed for this study. Spirometry was performed using handheld EasyOne spirometers^®^ (ndd Medizintechnik AG, Zurich, Switzerland), which was previously validated and used in a number of studies [4,10]. Key staff were trained in spirometry technics by a pulmonologist. Further on-site training was provided to the technical staff by trained principal investigators. Spirometry tests were performed according to ATS/ERS (European Respiratory Society) guidelines [9]. Recorded spirometry parameters were forced vital capacity (FVC), forced expiratory volume in 1 s (FEV1) and peak expiratory flow (PEF). A rigorous internal and external quality control process was established based on the spirometry guidelines developed for the TB Sequel study to identify usable curves for inclusion in the final analysis [7].

### 2.3. Data Analysis

Tabulations were done to summarize participants’ characteristics. Different modelling strategies were followed to generate new Mozambican prediction equations based on the measured spirometry outcomes. FEV1 and FVC were modelled individually using regression models as reported in most studies, including sex, height and age as covariates [11,12]. As evolution with age follows a nonlinear trend, complex models like generalized additive models for location scale and shape (GAMLSS) were used by the GLI in previous studies with large sample size [13,14]. However, due to limited samples size with our data, generalized additive models (GAM) models with the increased complexity for smoothing the effect of age on the outcomes did not perform significantly better than multiple linear regression models. Thus, we finally used regression models including age, height and sex as covariates to predict values for FVC and FEV1. We performed a sensitivity analysis using multiple cross-validation methods: (1) random sample of 2/3rd of the data as training data set, followed by the remaining 1/3rd as the testing data set; (2) leave one out cross validation (LOOCV) or the Jack Knife estimator; (3) k = 10 fold cross-validation; and (4) k = 10 fold cross-validation with 5 repetitions, to examine the predictive accuracy of our model. The newly generated prediction models were evaluated for predictive efficacy using the measures of root mean square errors (RMSE), mean absolute error (MAE) and R2 (describing the squared correlation between observed and predicted in the test data set) [15]. Further, the stability of regression equations was evaluated by comparing the regression estimates to the non-parametric bootstrap estimates and the corresponding 95% confidence intervals based on 10,000 resamples from our observed data [16,17,18]. Finally, we assessed the performance of the newly derived Mozambican prediction equation as well as the differences in outcomes using the South African Black and GLI—Others based equations on a spirometry data set from a recently described (post-) TB cohort [6].

The newly generated Mozambican prediction equations were also used to analyze the lung function of individual participants in this study. We calculated a z-score (otherwise known as standardized residual score, or SRS) for the measured FVC, FEV1 and FEV1/FVC ratio in each participant to define how many standard deviations the measured value was away from the predicted value. In line with most recent guidelines, the lower limit of normality (LLN) for each spirometric parameter is represented by a z-score of −1.64, which is equal to the lower 5th percentile of the standard population. That means, participants with a z-score of −1.64 and lower for FVC, FEV1 or FEV1/FVC have lung function parameters of below the 5th percentile of predicted (=LLN) and, thus, are diagnosed with restrictive or obstructive lung function impairment, respectively [19,20,21]. Severity grading was done as follows: (1) mild impairment: FVC or FEV1/FVC > 85% LLN; (2) moderate: FVC or FEV1/FVC 55–85% of LLN; (3) severe: FVC or FEV1/FVC < 55% of LLN [21]. Resulting differences in ventilation patterns and severity gradings based on the use of the different prediction equations (new Mozambican, GLI—using category “Others”, and South African references for the black population) were also assessed [2,22]. In the absence of a South African prediction equation for the FEV1/FVC ratio, we used the GLI equation [2].

### 2.4. Ethics Approval and Consent to Participate

The study was approved by the Comité Nacional de Bioética para Saúde (CNBS, reference 449/CNBS/16). Written informed consent was obtained from all study participants.

## 3. Results

### 3.1. Characteristics of the Study Participants

A total of 212 subjects were recruited in the study, of whom 155 had usable spirometry results and were included in the final analysis. Table 1 shows the characteristics of the participants. The mean age was 35.20 years (SD 10.99) and almost 60% of participants were female (93 of 155; 59.35%). The majority of participants (136 of 155; 87.74%) have never smoked and a relatively high proportion (37.96%) reported as HIV-positive.

### 3.2. Mozambican Spirometric Reference Equations

We modelled regression equations for each spirometric parameter (FVC, FEV1 and FEV1/FVC ratio) based on the spirometry values obtained from our Mozambican study population, as shown in Table 2. Age, height and sex were covariates for the prediction of FVC, and additionally, the influence of height differed by sex for FEV1, but not for FVC (Table 2 and Figure 1). We could further show that FVC and FEV1 were highly correlated (Appendix A). The ratio for FEV1/FVC was dependent on the age when we observed a decreasing trend in the ratio with an increase in age (Table 2 and Appendix A). Here, no difference was observed among males and females. A comparison of the final model (most parsimonious) with others—more complex (more parameter) and simpler (less parameter) models, arranged hierarchically, and their corresponding likelihood ratio test results are described in Appendix A.

Legend: Lines (LOESS (locally estimated scatterplot smoothing) fit) for FEV1 and FVC, stratified by sex, show predicted values and scatter plots show the spread of the actual data. Fitted lines show the same development over age, with different intercepts for males vs. females. However, for the association with height a difference in slopes for males vs. females is visible. The relationship for the ratio of FEV1/FVC with age, sex and height is shown in the Appendix A.

### 3.3. Lung Impairment in Mozambican Sample: Type and Severity Grading

The newly generated equations were used to calculate the LLN and a z-score for the measured values for FVC, and FEV1 and calculated FEV1/FVC ratio in each study participant.

Table 3 and Appendix A show the lung function of our participants with regards to impairment type and severity grade. Out of the 155 included participants, 16 (10.3%, 95% CI: 6.02% to 16.22%) had abnormal lung function, with nine having an FEV1/FVC ratio below the LLN (obstruction) and seven having FVC values below the LLN (restriction). With regard to severity grading, apart from one subject with moderate obstruction, all other 15 participants with abnormal lung function had only mild impairment (Table 3 and Appendix A). None of the risk factors listed in Table 1 was significantly associated with any spirometric parameter.

### 3.4. Comparison of Spirometry Results Based on Different Reference Standards

We assessed the lung function of our study participants by using different reference standards (Mozambican—Local, South African—Black and GLI—Others), and then compared the outcomes. Figure 2 shows the density distribution of the z-scores for FEV1 and FVC for each of the three different reference standards when applied to our study population. Compared to the newly created Mozambican reference standard, a clear shift to lower z-scores for FEV1 and FVC can be observed when the South African and GLI standards are applied to the study sample, resulting in a higher proportion of subjects with abnormal lung function. Complementary to the density distribution of z-scores, we also see a much better fit for the observed data of FVC and FEV1 (in liters) with the Mozambican predictions, while there is a clear shift to and higher predicted values for FEV1 and FVC on average when using the South African and GLI reference standards (Appendix A). No difference was observed for the FEV1/FVC ratio across the different standards. In line with that, the difference in z-scores for FVC between Mozambican prediction equations and the other two predictions was on average one standard deviation (z-score difference = 0.9). For FEV1 z-scores, the overall difference compared to GLI prediction was 1.2 standard deviations but only 0.4 standard deviations compared to South African prediction (Appendix A).

Legend: Compared to the newly created Mozambican equations, a clear shift to lower z-scores for FEV1 and FVC can be observed if the South African and GLI equations are applied to the study sample, resulting in a higher proportion of subjects with abnormal lung function.

These findings are supported by the observed numbers of participants with abnormal lung function and the corresponding severity grading, depending on the different reference standard used for the studied population (Table 3 and Appendix A). Using GLI and South African standards results in a relevantly higher number of subjects with pulmonary restriction (= FVC values below LLN: 21.3% for GLI; 18.7% for South African) compared to applying Mozambican prediction equations (4.5% of FVC values with a z-score of below −1.64). On average, z-scores for FEV1 retrieved from GLI and South African standards equations were also lower compared to z-scores based on Mozambican predictions, there were also higher proportions of subjects with obstruction, which corresponds to an abnormal FEV1/FVC ratio (8.4% for GLI and South African; 5.8% for Mozambique) (Table 3 and Appendix A). The number of subjects classified as having abnormal lung function would increase alongside the number of subjects diagnosed with moderate and even severe impairment if non-Mozambican prediction equations were used (Table 3 and Appendix A). The fact that the GLI derived z-scores for FEV1 were lower on average than those for FVC (Appendix A) resulted in five (5/155 = 3.2%) subjects, who would be diagnosed with severe obstructive impairment (three of them with mixed impairment) compared to none if the Mozambican equations would be used as a reference.

### 3.5. Validation of Models

To validate our modelled prediction equations, we performed several sensitivity analyses (Appendix A). In most scenarios, the assessed parameters (RMSE, MAE and R2) performed better for the models built on the Mozambique data than South African or GLI prediction equation models. Additionally, in terms of stability of estimates, the bootstrapped confidence intervals from the Mozambican data were in alignment with the estimated regression coefficients estimated by multiple linear regression methods (Appendix A).

As another test for our prediction model, we applied the new equations as well as GLI and South African standards to the spirometry data of a (post-) TB cohort [6]. As expected, for all reference standards, we observed the improvement of lung function under treatment, with best lung function results at 52 weeks after TB diagnosis and treatment start. However, the comparison of z-scores of the different standards shows that less impairment is described by the local Mozambican standard compared to South African and GLI equations, suggesting overall better fits for the Mozambican standard (Appendix A).

## 4. Discussion

Ethnicity has been recognized to play a significant role in the variability of lung function, thus it is important to establish reference values relevant to the ethnic characteristics of the local population [23,24,25]. In this study, we have generated prediction equations for FVC and FEV1 as well as for the ratio of FVC/FEV1 based on lung function data from 155 healthy adults living in Maputo, Mozambique. Compared to the prediction equations from the neighboring country, South Africa, the estimated coefficients associated with age and height of the Mozambican formulas were mostly smaller, except height in FEV1 for males. The South African model regression parameters were not statistically significantly different from the local Mozambican population-based regression model; however, the implications on predictions were different. This resulted in lower predicted values for FEV1, FVC and the FEV1/FVC ratio and, thus, higher (more normal) z-scores for measured values of study participants. Even larger differences were observed in a comparison of predicted values and z-scores based on the new Mozambican prediction equations versus GLI equations. This means, in the absence of the newly generated Mozambican standard, a relevant proportion of our asymptomatic study participants might have been misclassified and placed in abnormal lung function categories, including shifts into higher severity grades, which would be considered clinically relevant. Thus, these patients should be considerably limited in lung function and be symptomatic during routine activities. However, this was denied by all participants at the recruitment. The same trend was observed when we applied our newly derived Mozambican equation, GLI and South African equations to our dataset derived from the (post-) TB cohort in Maputo, Mozambique. As the differences were substantial in some subjects, our findings are relevant for clinical practice. They suggest that individuals from certain ethnic groups might be incorrectly diagnosed and treated for pulmonary conditions due to the application of an inappropriate reference standard to their spirometry results [25]. However, we cannot exclude that lung damage is prevalent in a certain proportion of our clinically asymptomatic study participants. In fact, about 10% of our study population was diagnosed with mild spirometric abnormalities based on the newly generated Mozambican standard. Similar to other spirometric studies with healthy adults included in GLI prediction equations calculations, data on exclusion criteria such as acute or chronic respiratory symptoms and medical history of lung conditions were collected with a standardized and validated questionnaire in our study. Thus, “healthy” volunteers are often equivalent to non-symptomatic (asymptomatic) adults or children, which, however, does not necessarily exclude abnormal results for physiological lung function testing. Further, our study population was recruited in a poor urban area of Maputo with a high prevalence of risk factors for lung health, such as indoor and environmental air pollution, smoking habits or recurrent respiratory infections in childhood, which might explain the mild pathology found in the participants of this study. Interestingly, findings on lung impairment are rarely reported from other spirometric surveys with asymptomatic adults, as they are mostly limited to the description of the newly derived equations and associated methodology. However, regardless from what asymptomatic (healthy) cohort spirometric values were taken, due to the underlying basic statistical assumptions (normally distributed), the resulting modelled prediction equations will always lead to the diagnosis of abnormally low values in some of the survey participants, usually in the 5% with the lowest and highest values. For analysis of spirometry data, the lowest 5% are considered as abnormal, according to internationally accepted conventions [20]. In order to establish a spirometric reference standard, which is generalizable to a broader Mozambican population, the lung function of more volunteers with a greater age span and from different socio-economic backgrounds would need to be analyzed.

There are a number of limitations of our study that may also render the newly generated prediction equations not generalizable to the broader population of Maputo or Mozambique. Firstly, our study has a small sample size. As we employed a convenient sampling strategy to generate a local spirometric reference from the general population from which we recruited a TB-cohort [6], we were able to recruit only 212 volunteers in the temporal context of the main study. Recently, Quanjer et al. 2012 suggested that in order to validate reference equations for spirometry data, a sample of 300 healthy subjects would be more favorable [14]. Simulation studies in the publication by Austin and Steyerberg, 2015 [26] suggested, that only two subjects per variable are needed for an adequate estimation of coefficients in linear regression models. However, testing for the predictive accuracy of the regression model would require an effectively larger sample size to have separate training and testing data, which we lacked by design. Nevertheless, we still tried to evaluate the predictive accuracy of our model using sensitivity analysis by fitting the regression models on subsamples of the observed data and testing on the remaining data. Finally, there is a huge variety in sample sizes of spirometric data sets, which were included in GLI multi-ethnic reference values calculations: from 108 participants in Ben Saad et al., 2008—Tunis [27] to 5315 in Pérez-Padilla et al., 2006—five Latin American countries [28], indicating that also smaller data sets, which were collected according to ATS/ERS and GLI recommendations, are valuable and could provide important evidence for the generation of prediction equations, that are more representative for specific populations, e.g., in Sub-Saharan Africa. Secondly, our study could only include participants older than 18 years, despite the current trend of including people of all ages in surveys aiming for the generation of spirometric reference equations [2].

Finally, 57 out of 212 participants (26.9%) did not have a valid spirometry result, either due to their inability to pass validity criteria or due to the presence of contraindications such as high blood pressure before the test. Those had to be excluded from the final analysis. There were no statistically significant differences between the included and non-included subjects for assessed risk factors including age, sex, height, weight, smoking status and self-reported HIV status. However, other risk factors than those measured might have been differently distributed among the two groups and, thus, could have introduced bias.

Acknowledging the recent calls for the adoption of the GLI reference values in clinical practice worldwide [29], our study is still very relevant. In the Mozambican context, our data suggest that the GLI equations may not apply to the Mozambican population, mainly because they did not include data from this country, and that the newly generated prediction equations may be an important reference for local clinicians and researchers to critically appraise the lung function of their patients with and without respiratory symptoms to avoid diagnostic errors.

## 5. Conclusions

To our knowledge, this was the first study to obtain local population spirometric equations in Maputo, Mozambique. Our results support the assumption that the GLI-based reference standards (category “Others”) may not be appropriate for our population, because too few data from African populations were included in the equation modelling process. This may result in diagnostic errors in asymptomatic persons as well as in patients with respiratory symptoms. Therefore, the study results will contribute, in addition to other published standards such as GLI, a valuable comparison for future analysis of spirometry results from patients with pulmonary TB or other lung diseases that were recruited into different clinical studies from a similar urban environment. This study was performed in line with GLI and ERS Task Force recommendations that are calling for data collections in non-Caucasian, particularly African and Latin American, populations, including ethnic minorities [2].

## Figures and Tables

**Figure 1 ijerph-17-04535-f001:**
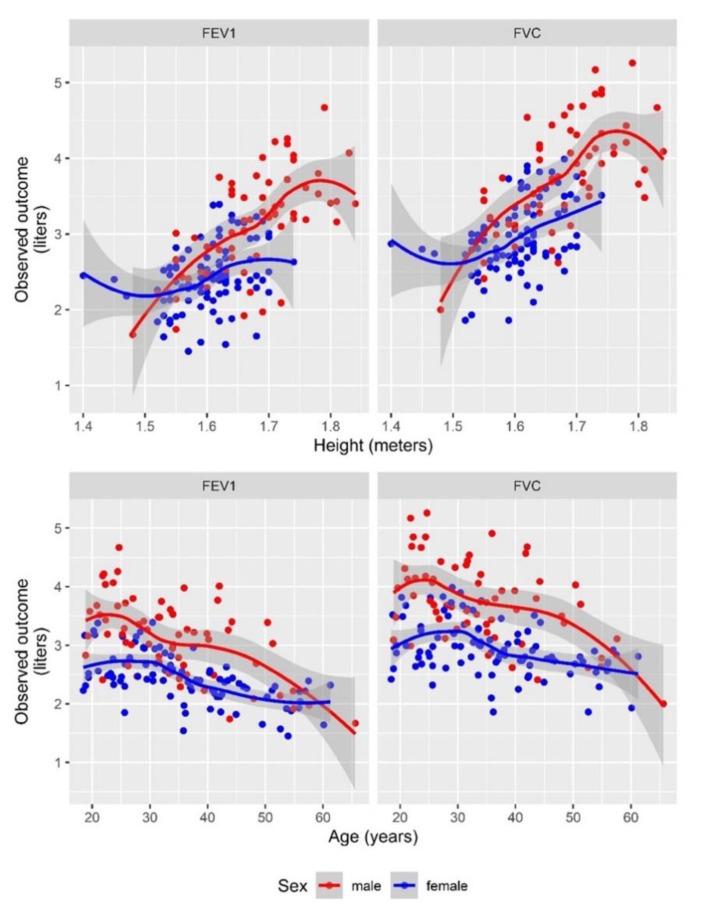
Association of FVC and FEV1 values with height and age and according to sex.

**Figure 2 ijerph-17-04535-f002:**
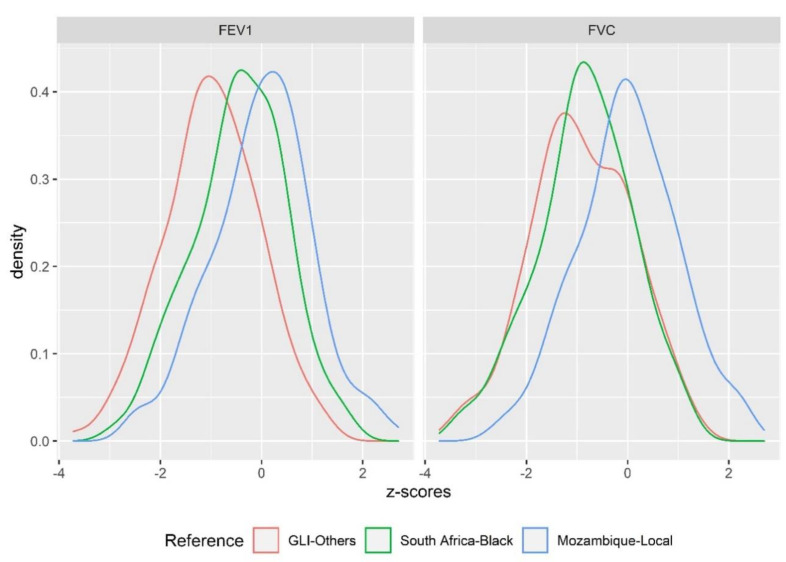
The density distribution of z-scores from the Mozambican sample based on three different reference standards (prediction equations).

**Table 1 ijerph-17-04535-t001:** Anthropometric, demographic and spirometric characteristics of the participants.

Characteristics	Male (*n* = 63)	Female (*n* = 92)	Total (*n* = 155)
Age (years)	33.83 (SD 10.74)	36.13 (SD 11.12)	35.20 (SD 10.99)
Height (meters)	1.67 (SD 0.08)	1.60 (SD 0.06)	1.63 (SD 0.08)
Weight (kg s)	65.52 (SD 9.80)	69.24 (SD 15.68)	67.73 (SD 13.69)
BMI (kg/sq. meters)	23.43 (SD 3.70)	26.95 (SD 5.60)	25.52 (SD 5.20)
**Age group**			
<30 years	27 (42.86%)	30 (32.61%)	57 (36.77%)
30–40 years	20 (31.75%)	28 (30.43%)	48 (30.97%)
≥40 years	16 (25.39%)	34 (39.96%)	50 (32.26%)
**BMI Class ***			
Underweight	2 (3.17%)	2 (2.17%)	4 (2.58%)
Normal	45 (71.43%)	38 (41.30%)	83 (53.55%)
Overweight	11 (17.46%)	28 (30.43%)	39 (25.16%)
Obese	5 (7.94%)	24 (26.09%)	29 (18.71%)
**Smoking**			
Never Smoked	49 (77.78%)	87 (94.57%)	136 (87.74%)
Past Smoker	8 (12.70%)	5 (5.43%)	13 (8.39%)
Current Smoker	6 (9.52%)	0 (0%)	6 (3.87%)
**Marital Status**			
Single	25 (39.68%)	39 (42.39%)	64 (41.29%)
Married	11 (17.46%)	11 (11.96%)	22 (14.19%)
Living with spouse/partner	25 (39.68%)	37 (40.22%)	62 (40.00%)
Widowed	2 (3.17%)	5 (5.43%)	7 (4.52%)
**Education**			
No formal education	0 (0%)	2 (2.17%)	2 (1.29%)
Grades 1–5	5 (7.94%)	24 (26.09%)	29 (18.71%)
Grades 6–10	27 (42.86%)	43 (46.74%)	70 (45.16%)
Grades 11–12	17 (26.98%)	20 (21.74%)	37 (23.87%)
Vocational	8 (12.70%)	2 (2.17%)	10 (6.45%)
University	6 (9.52%)	1 (1.09%)	7 (4.52%)
**HIV status** (self-reported) (*n* = 108, 47 missing observations)			
Negative	27 (72.97%)	40 (56.34%)	67 (62.04%)
Positive	10 (27.03%)	31 (43.66%)	41 (37.96%)
**Worked in Mines**			
No	61 (96.83%)	92 (100%)	153 (98.71%)
Yes	2 (3.17%)	0 (0%)	2 (1.29%)
**Spirometric Parameters ****			
FVC (L)	3.77 (SD 0.69)	2.94 (SD 0.46)	3.28 (SD 0.70)
FVC (% of predicted)	90.82 (SD 11.43)	88.62 (SD 11.76)	89.51 (SD 11.65)
FEV1 (L)	3.12 (SD 0.67)	2.43 (SD 0.42)	2.71 (SD 0.63)
FEV1 (% of predicted)	91.28 (SD 13.69)	95.53 (SD 13.02)	93.80 (SD 13.41)
FEV1/FVC ratio	0.83 (SD 0.06)	0.83 (SD 0.06)	0.83 (SD 0.06)

Legend: * BMI according to World Health Organization (WHO) classification; ** predicted FVC and FEV1 based on South African reference standards [22].

**Table 2 ijerph-17-04535-t002:** New spirometric prediction equations obtained from the study sample in comparison to the South African equations.

Outcome (Sex Specific)	South African (Black) Population	Mozambique (Local) Population
FVC (Males)	−3.08 − 0.024 × Age + 4.8 × Height; RSS = 0.54	−2.271 − 0.019 × Age + 3.989 × Height; RSS = 0.43; adj Rsquare = 0.61
FVC (Females)	−3.04 − 0.023 × Age + 4.5 × Height; RSS = 0.41	−2.761 − 0.019 × Age + 3.989 × Height; RSS = 0.43; adj Rsquare = 0.61
FEV1 (Males)	−0.54 − 0.027 × Age + 2.9 × Height; RSS = 0.46	−3.504 − 0.023 × Age + 4.426 × Height; RSS = 0.37; adj Rsquare = 0.65
FEV1 (Females)	−1.87 − 0.028 × Age + 3.4 × Height; RSS = 0.39	−0.170 − 0.023 × Age + 2.150 × Height; RSS = 0.37; adj Rsquare = 0.65
Ratio FEV1/FVC (Not sex specific)	-	0.921 − 0.0027 × Age; RSS = 0.06; adj Rsquare = 0.22

Legend: The regression estimates are smaller in magnitude for Mozambican compared to South African equations, however, the direction of association is the same. We modelled the ratio of forced expiratory volume in 1 s/forced vital capacity (FEV1/FVC). The adjusted, comparably low value for R square (=0.22) indicates that age is only explaining 22% of the observed variability in the ratio of FEV1 and FVC. For the individual outcomes for FEV1 and FVC, both values for R square were higher than 0.6 and hence more than 60% of the variation observed in FEV1 and FVC are explained by the covariates in the regression equation. The Global Lung Initiative (GLI) equations used in this article are based on generalized additive models for location scale and shape (GAMLSS) models and, hence, the regression estimates are not directly comparable and therefore not included in Table 2.

**Table 3 ijerph-17-04535-t003:** Comparison of outcome categories using Mozambican prediction equations versus South African and GLI equations.

Impairment Type and Severity Grading, N = 155	Mozambique—Local % (n/N)	GLI—Others % (n/N)	South Africa—Black % (n/N)
Normal	89.7 (139/155)	72.9 (113/155)	74.8 (116/155)
Obstruction—Mild	5.2 (8/155)	2.6 (4/155)	5.8 (9/155)
Obstruction—Moderate	0.6 (1/155)	1.9 (3/155)	0.6 (1/155)
Obstruction—Severe	0.0 (0/155)	1.3 (2/155)	0.0 (0/155)
Restriction—Mild	4.5 (7/155)	15.5 (24/155)	12.9 (20/155)
Restriction—Moderate	0.0 (0/155)	3.2 (5/155)	3.9 (6/155)
Restriction—Severe	0.0 (0/155)	0.0 (0/155)	0.0 (0/155)
Mixed—Mild	0.0 (0/155)	0.0 (0/155)	0.0 (0/155)
Mixed—Moderate	0.0 (0/155)	0.6 (1/155)	1.3 (2/155)
Mixed—Severe	0.0 (0/155)	1.9 (3/155)	0.6 (1/155)

Legend: While only 16 (10.3%) subjects had abnormal lung function according to the Mozambican reference standard, 42 (27.1%) and 39 (25.2%) subjects had lung impairment if GLI and South African standards, respectively, had been applied. The greatest discrepancies among the three reference standards are present in the restriction- and mixed- categories as well as in severity categories moderate and severe.

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
