# Peer review of "Lung Function Testing and Prediction Equations in Adult Population from Maputo, Mozambique"

_ijerph, 2020, doi:10.3390/ijerph17124535_

Round 1

Reviewer 1 Report

In this paper, the authors attempt to make lung function prediction equations for Mozambique using 155 healthy adults.

Major Issue:

I think the major issue revolves around the sample, 1) is it enough, and 2) is it healthy. 

  1. It’s not clear if they really have enough people to do this. This is briefly touched on in the Discussion where they mention that 300 subjects are recommended. 
  2. There’s a bit of chicken-and-egg problem because it’s hard to say if the cohort is healthy, or if it’s got 25% impaired spirometry, as indicated by the GLI or South African equations.
    When the percentages of impairment were compared for the GLI, Mozambique and SA equations, the Moz equations classify fewer people as impaired. This could also be because the Study Population has a high percentage of individuals with impaired lung function, and basing a system of measurements on these individuals includes a substantial portion of people with impaired lung function, which brings the averages down.  This would then result in lower predictions of average lung function according to the new Moz equations, and would account for the lower proportion of people classified as having impairment under the Moz equations.

I think these are both probably difficult to remedy directly, but the paper would benefit from some expanded discussion on these points.

Minor Issues

Define GAMLSS.

The methods briefly mention splitting the data into training and testing; which I think is just because they’re not actually discussing how they did the training/testing split, the cross validation, or the bootstrapping, which are all mentioned in the Supplement.  This could be touched up a little bit so it makes more sense in context.

Line 92: what’s the rationale for this LLN definition?  Isn’t the LLN usually the fifth percentile in COPD?  Also -1.64 z-scores seems remarkably uninterpretable.

Again, the LLN is supposed to be based on healthy subjects; it seems like they’re including a bunch of subjects with mildly pathological lung function.

The results of the Cross-validation or the Bootstrapping or the Hold Out split of Training/Testing data should be presented in the results, not the discussion.

I don’t know if this is obvious, but it should probably be stated if the South African reference equations are based on South Africans of European or African heritage.

Discussion: To what extent is this population representative of the wider African population?

Author Response

Thank you! Please see the PDF attachment.

Reviewer 2 Report

I thank the authors for undertaking this study. Spirometry reference values are a critical need for Africa

I have a few concerns

  1. The sampling is not clear. In surveys to determine reference values effort should be undertaken to ensure the studied population represents the normal. It is not clear if this was the case. I see smokers were included according to table 1. This should be clarified
  2. The sample size is very small; was a sample size calculated
  3. Looks participants were taken from one urban sample. Does this represent the entire country
  4. The paper could actually emphasize the lung function abnormalities found and discuss them in their own right

Author Response

(The authors gave the same response as above.)

Reviewer 3 Report

The paper presents a lung function prediction equation for local adult population of Maputo, Mozambique and compares this equation with equation derived by the Global Lung Initiative, and equation derived from the larger South African Population. The premise of the paper is that the lack of data from the African region makes the lung function prediction equation derived by either GLI or SA less applicable to the local population which differs considerably in their ethnic origin and anthropometric characteristics, and therefore, a separate equation should be considered from the local population. The author argue that the equation derived from the local population is indeed more accurate.   One of the major limitation of this work is the rather small number of samples, i.e., 155. The authors mention this in the discussion section as well, i.e., the equation derived in the paper might not be applicable to a more general population even from the same region. Additionally, the paper relies on 'convenience sampling' which is perhaps not the most concrete way to build a reliable model since it is not appropriate representative of the population. A larger sample size would definitely increase the value of this work.  
  • The authors conclude that the fitted model assessed 16/155 participants as having abnormal lung function. Are these 16 participants false positives given that the model should be fitted on normal participants? Are the authors implying that this number (i.e., 16) should ideally be 0, and given it is lower than the same numbers given by the GLI and SA models, it shows that the locally fitted model is providing better insight?
  • It will be nice to see visually how poorly the GLI and SA equations perform on the dataset, for example, the authors might consider showing the observed and predicted values of FEV1 and FVC from these two approaches. They should deviate considerably from the diagonal.
  • Is there any particular reason the RSquare values between SA and local model differ significantly for males but not for females in Table 2? Are these results computed from a test set or using LOOCV? Can the authors clarify how the test set was built?
  • Can the authors provide the confidence interval from the bootstrapping in the main article? Do these values differ statistically from the SA model as reported in Table 2?

  • "Predictions for FVC, FEV1 and the FEV1/FVC ratio based on the Mozambique equations were lower than the South African and the GLI based predictions." This line is not clear to me. Does the author imply that the prediction errors are smaller or do they imply that the predictions themselves take smaller value. If it is the latter then what does exactly that means in this context, i.e., is it better and if so then how?
  • The curves presented in Figure 1 are nonlinear while the equations presented in Table 2 are linear. Are they both for the equations derived on the local population?
  • The resolution of the images are rather low.
  • What is the difference between Table 2 and 3?
  • Did the authors check the impact of the other predictors listed in Table 1, e.g., HIV and Worked in Mines on the FEV1/FVC values?
  • In Figure 2, were the z-score values computed on test data?

Author Response

(The authors gave the same response as above.)

Round 2

Reviewer 2 Report

I thank the authors for revising the manuscript. The weakness that remain are probably not fixable especially the issue of sample size. The participants with abnormal spirometry could dropped from the sample. This study should only help the authors to plan a bigger study. I am convinced with the response. 

Author Response

Dear Reviewer, first of all thank you very much for finding time and reviewing our paper. We are also thankful for accepting our explanations and corrections. We agree that, unfortunately, we cannot fix sample size number, however, we have tried to explain and justify it to our best knowledge and using previous evidence. We are in process of collecting more spirometry data from asymptomatic “healthy” volunteers, including children, in frame of our multi-center cohort – TB Sequel. We are ensuring the adequate sample size in this survey.

We would also like to emphasize at this point, that regardless from what asymptomatic (healthy) cohort spirometric values were taken, due to the underlying basic statistical assumptions (normally distributed), the resulting modelled prediction equations will always lead to the diagnosis of abnormally low values in some of the survey participants, usually in those 5% with the lowest and highest values. For analysis of spirometry data, the lowest 5% are considered as abnormal according to suggested conventions (ERS/ATS guidelines). As the assessment of lung function is based on three different parameters (prediction equations), namely FVC, FEV1 and FEV1/FVC-ratio, which, if too low, could lead to the diagnosis of an abnormal lung function, the observed finding of 10% of subjects with different types (obstruction and restriction, about 5% of each type) of abnormal lung function in our study (when applying Mozambican prediction equations) is still in the scope of what can be expected, especially as severity of impairment was only mild in all (but one) cases.

Reviewer 3 Report

I sincerely thank the authors for the elaborate feedback on my comments. I believe that the modifications have significantly improved the clarity and content of the paper.

I will recommend including in the paper that 'the SA model regression parameters are not statistically significantly different from the local Mozambican population regression model.'

I will also recommend including observed-versus-predicted values plot since Figure 3 shows only the marginal structure and not the joint which might reveal information about the difference of fit.

Author Response

I sincerely thank the authors for the elaborate feedback on my comments. I believe that the modifications have significantly improved the clarity and content of the paper.

Reply: Dear Reviewer, first of all thank you very much for finding time and reviewing our paper. We are also thankful for accepting our explanations and corrections.

I will recommend including in the paper that 'the SA model regression parameters are not statistically significantly different from the local Mozambican population regression model.

Reply: Thank you for the recommendation, we have included this statement in line 248-250.

I will also recommend including observed-versus-predicted values plot since Figure 3 shows only the marginal structure and not the joint which might reveal information about the difference of fit.

Reply: Thank you for this suggestion. We have substituted Figure 3 in the supplement with the observed-versus-predicted values plot as recommended and added explanatory legend. We have also adapted a description in the manuscript text, lines 189-191.